# FDG PET/CT as a Tool for Early Detection of Bleomycin-Induced Pulmonary Toxicity

**Hira Shaikh** [1,*] , **Zulfa Omer** [2] , **Roman A. Jandarov** [3] , **Morgan P. McBee** [4] , **Jennifer Scheler** [5] , **Bruce Mahoney** [5] **and Tahir Latif** [2]

1    Department of Hematology, Oncology and Bone Marrow Transplantation, University of Iowa Hospitals and Clinics, Iowa City, IA 52242, USA

2    Department of Hematology Oncology, University of Cincinnati Medical Center, Cincinnati, OH 45267, USA; omerza@ucmail.uc.edu (Z.O.)

3    Department of Environmental Health, University of Cincinnati, Cincinnati, OH 45267, USA

4    Department of Radiology and Radiological Science, Medical University of South Carolina, Charleston, SC 29425, USA

5    Department of Radiology, University of Cincinnati Medical Center, Cincinnati, OH 45267, USA

*    Correspondence: hiragshaikh@outlook.com

**Abstract:** Bleomycin-induced pulmonary toxicity (BPT) is a serious and potentially fatal complication of bleomycin, a key component of Hodgkin lymphoma (HL) treatment. Before ours, only one published study evaluated the predictability of 18F-FDG-PET/CT for the early diagnosis of BPT. In this retrospective cohort study, 18F-FDG-PET/CT scans of adult HL patients treated with bleomycin at an urban academic center over five years were assessed by radiologists blinded to the clinical information, and scans were correlated with clinical BPT. We found 11 HL patients with 54 interim or end-of-treatment 18F-FDG-PET/CT scans who had received bleomycin. Five of the eleven (5/11, 45%) patients had radiographic changes in PET/CT and developed clinical BPT. Patients with clinical BPT had higher FDG uptake in lungs compared to those who did not ($SUV_{max}$ mean 2.66 (CI 1.8–3.7) vs. 0.86 (CI 0.4–1.9), Mann–Whitney U test, $p < 0.05$). In a separate cohort analysis, we compared HL patients with clinical BPT (9/25, 36%) and without clinical BPT (16/25, 64%) to assess potential risk factors. Low hemoglobin ($p = 0.037$) and high ESR values ($p = 0.0289$) were associated with clinical BPT. Furthermore, gender, stage, histology, prior lung radiation, G-CSF, or steroids did not significantly confer a higher risk of BPT. 18F-FDG-PET/CT imaging, which is routinely used to assess treatment response in HL, is useful for early detection of BPT, which can have high mortality and morbidity.

**Keywords:** bleomycin; pulmonary toxicity; Hodgkin lymphoma; FDG-PET/CT





## 1. Introduction

Pulmonary toxicity is the most serious adverse event associated with bleomycin. This is known to occur in around 10–53% of patients treated with bleomycin [1–4], with higher frequency associated with advanced age (>40–65 years) [5–7], higher bleomycin cumulative dose [5,7,8], and prior treatment with thoracic radiation [9]. However, none of these factors have been validated as reliable predictors of the risk of bleomycin pulmonary toxicity (BPT) in an individual patient. Mortality rate with BPT can be as high as 10–20% [10].

BPT is thought to be related to the lack of bleomycin-inactivating enzyme (bleomycin hydrolase) in the lungs [8]. Pathogenesis involves bleomycin-induced release of cytokines and free radicals that cause endothelial damage, followed by influx of inflammatory cells into lungs, and subsequently lead to the activation of fibroblasts, resulting in fibrosis [11–13]. This is a serious concern associated with treatment of Hodgkin lymphoma (HL) and involves ABVD (adriamycin, bleomycin, vincristine, and dacarbazine).

The diagnosis of BPT is usually made based on the combination of a compatible clinical and radiological pattern or asymptomatic decline in the diffusing capacity of lungs for carbon monoxide (DLCO) by 25% during treatment with bleomycin and the exclusion of infection, radiation related changes, or pulmonary involvement in underlying malignancy [14]. This becomes challenging in the absence of pathognomonic features, and diagnosis can be delayed by overlap with other conditions often encountered in cancer patients that can mimic similar presentation, such as infections, pulmonary metastasis, and lymphangitic carcinoma. Lung biopsy is done if the diagnosis remains unclear after other etiologies are ruled out. However, histopathologic findings of BPT are nonspecific and require correlation with clinical findings. Due to high morbidity of this condition, it is essential that bleomycin be discontinued if there is a strong concern of BPT. There is some evidence of systemic glucocorticoids being effective, however, their use is typically limited to symptomatic patients with acute decline in DLCO [15].

Integrated 18F-fluorodeoxyglucose (FDG) positron emission tomography/computed tomography (PET/CT) is essential in diagnosis of HL and is a major prognostic factor of disease control after two courses of ABVD (PET-2) [16,17]. There have been case reports of BPT diagnosed with FDG PET/CT [18–20]. FDG PET/CT findings of BPT have reported diffuse or focal regions of ground-glass opacities associated with increased pulmonary FDG uptake [21,22]. In a study done by Hollander et al., 68% of patients had radiological changes suspicious of BPT, and most pulmonary changes were multifocal and found in the basal parts of the lungs [23]. Some radiographic findings of PET were suspicious of BPT but without any clinical manifestations, and the condition resolved with stopping of bleomycin administration [19]. Thus, this suggested 18F-FDG PET/CT to be instrumental in diagnosis of early, preclinical BPT and can provide an opportunity for early cessation of bleomycin. However, only one study formally studied this condition as per our literature search [24]. Our study assessed whether 18F-FDG PET/CT findings correlate with the onset of BPT and can help in early diagnosis.

## 2. Results

### 2.1. Cohort 1—Clinical BPT and 18F-FDG PET/CT Findings

From January 2012 to December 2016, 11 HL patients (age > 18 years, treated with ABVD) with 54 total 18F-FDG PET/CT scans were included in the study. These included interim, restaging, and end-of-treatment scans. Initial diagnostic or staging scans were excluded for BPT assessment as these were performed prior to bleomycin administration; however, these were used for comparison. Median number of ABVD cycles received was 4 (range 3–6). Five patients received radiation therapy for HL, while data was unavailable for one patient who did not follow-up after referral for stem cell transplant (details in Table 1).

**Table 1.** Radiation therapy for treatment of Hodgkin lymphoma in cohort 1.

| Patient No. | Radiation Field | Radiation Dose (Gy) |
|:---:|:---:|:---:|
| 2 | Mediastinum and bilateral supraclavicular area | 30 |
| 5 | Whole lung irradiation | 10.5 * |
| 9 | Mediastinum | 30.6 |
| 10 | Right neck and retrosternal nodes | 27 |
| 11 | Mantle field | 30.6 |

* Completed 7 of 10 fractions, due to non-compliance.

Five of the eleven (45%) patients had radiographic changes in PET/CT and developed clinical BPT (Table 2). Among these, two patients with BPT had diffuse ground-glass opacities with increased FDG uptake (Figure 1), while three patients had focal areas of consolidation or ground-glass opacities with increased FDG uptake (Figure 2). One patient had diffuse ground-glass opacities in lungs with increased FDG uptake (maximum standardized uptake values ($SUV_{max}$) 1.8) but did not develop clinical BPT, that is, the patient

did not exhibit respiratory symptoms or qualifying changes in DLCO. Of note, this patient did have a pulmonary function test (PFT) abnormality post-treatment (mildly decreased diffusion capacity and forced vital capacity). The other five patients had no signs of toxicity on imaging (no increased FDG uptake within the lungs on PET and no ground-glass or consolidative opacities on CT) and did not develop clinical BPT (Table 2).

**Table 2.** Demographics and $SUV_{max}$ of patients with 18F-FDG-PET/CT scans evaluated for BPT.

| | **Negative for Clinical Toxicity** | **Positive for Clinical Toxicity** |
|---|---|---|
| N | 6 | 5 |
| Age (years) | $40.7 \pm 15.4$ | $40.6 \pm 20.1$ |
| Gender (Males/Females) | 5/1 | 3/2 |
| Mean $SUV_{max}$ of Lungs | 0.86 (CI 0.4–1.9) | 2.66 (CI 1.8–3.7) |

**FUSED BEFORE**　　　　　　**FUSED TOXICITY**

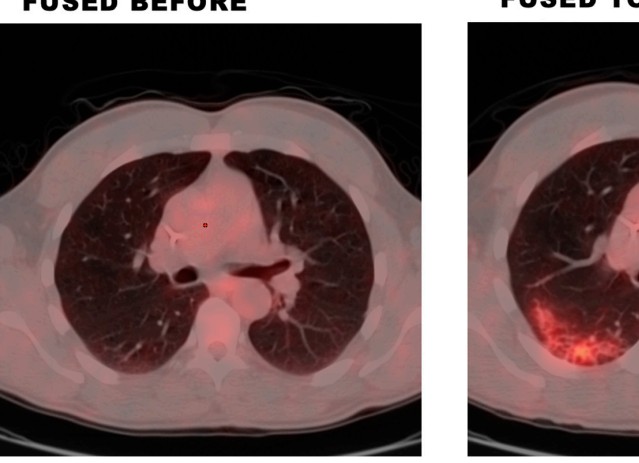

**Figure 1.** 18F-FDG-PET/CT showing diffuse ground-glass opacities with increased FDG uptake in a patient who had clinical BPT.

**FUSED BEFORE**　　　　　　**FUSED TOXICITY**

**Figure 2.** 18F-FDG-PET/CT exhibiting focal areas of consolidation or ground-glass opacity with increased FDG uptake in a patient who had clinical BPT.

The lungs of patients with clinical BPT had mean $SUV_{max}$ of 2.66 (CI 1.8–3.7), while lungs of those who did not develop BPT had mean $SUV_{max}$ of 0.86 (CI 0.4–1.9) (Figure 3). Mean $SUV_{max}$ was statistically higher for patients who developed clinical BPT ($p < 0.05$).

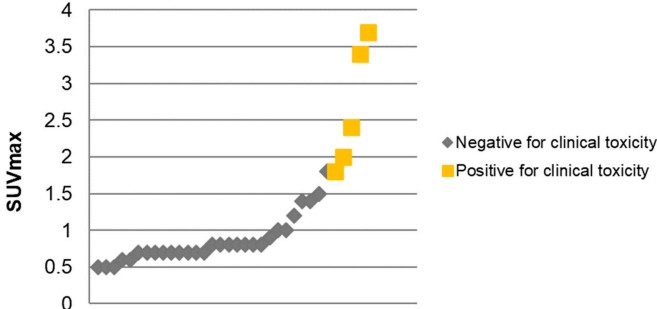

**Figure 3.** $SUV_{max}$ of lungs in 18F-FDG-PET, with and without clinical BPT. Patients with clinical BPT exhibited higher mean maximum standardized uptake values ($SUV_{max}$) of 2.66 (CI 1.8–3.7) than those who did not develop BPT and exhibited mean $SUV_{max}$ of 0.86 (CI 0.4–1.9); these were assessed using Mann–Whitney U test.

## 2.2. Cohort 2—Demographics and Disease Characteristics

A separate search of EMR for all HL patients (age > 18 years), regardless of PET/CT availability, treated with ABVD from January 2012 to December 2016 yielded 26 patients (Table 3). One patient was excluded due to incomplete records (received treatment outside of institution). Majority were males (72%, n = 18/25). The average age at diagnosis was 37 years (range 19–68). Most had an Eastern Cooperative Oncology Group (ECOG) performance status (PS) of 0 at diagnosis (65%, n = 15/23). Among patients who had staging data in the diagnosis available group (n = 24), the majority were stage II (54%, n = 13/24) followed by stage III (25%, n = 6/24). Most common type of HL was nodular sclerosing (56%, n = 14/25) followed by classical type (32%, n = 8/25). A small number of patients had bulky disease (12.5%, n = 3/24; data not available for one patient), while 52% (n = 13/25) had B symptoms at diagnosis.

**Table 3.** Demographics and disease characteristics.

| Demographics | | |
|---|---|---|
| Age, median (range) | | 30 (19–68) years |
| Males, n (%) | | 18/25 (72%) |
| ECOG PS [a], n (%) | 0 | 15/23 (65%) |
| | 1 | 8/23 (35%) |
| DISEASE CHARACTERISTICS | | |
| Bulky disease [a], n (%) | | 3/24 (12.5%) |
| B symptoms, n (%) | | 13/25 (52%) |
| Stage [a], n (%) | I | 1/24 (4.17%) |
| | II | 13/24 (54.17%) |
| | III | 6/24 (25%) |
| | IV | 4/24 (16.67%) |
| Histology, n (%) | Classical | 8/25 (32%) |
| | Nodular lymphocyte predominant | 2/25 (8%) |
| | Nodular sclerosing | 14/25 (56%) |
| | Nodular sclerosing/Mixed cellularity overlap | 1/25 (4%) |
| Albumin, median (range) | | 4.1 (3–4.9) g/dL |
| Hemoglobin, median (range) | | 12.5 (6.9–16.3) mg/dL |
| ESR, median (range) | | 38 (2–106) mm/h |
| ALC, median (range) | | 1495.5 (1.69–2292)/µL |

[a] Data missing for some patients as indicated in the total number. ALC—Absolute lymphocyte count; ECOG PS—Eastern Cooperative Oncology Group Performance Status; ESR—Erythrocyte sedimentation rate.

## 2.3. Cohort 2—Clinical BPT

In the total cohort, nine patients (9/25, 36%) developed clinical BPT. The median time from start of treatment with bleomycin to the onset of BPT was 3 months (range 2–11 months). One patient developed respiratory failure requiring mechanical ventilation and died, another transferred to a hospice, while the rest did not need respiratory support and were alive at the time of analysis.

## 2.4. Cohort 2—Risk Factors of Clinical BPT

Three patients had prior steroid exposure and all of them developed clinical BPT (Table 4). Three had prior growth factor colony stimulating factor (G-CSF) exposure, with one developing clinical BPT. Four patients had prior lung radiation, and one of these was diagnosed with BPT. Average cumulative bleomycin dose was 154 units (60–252).

**Table 4.** Clinical BPT classified by patient and disease characteristics.

| At Diagnosis | | N (%) | Clinical BPT | No Clinical BPT | *p* Value |
|---|---|---|---|---|---|
| Gender | Females | 7/25 (28%) | 3 (42.86%) | 4 (57.14%) | 1 |
| | Males | 18/25 (72%) | 6 (37.5%) | 10 (62.5%) | |
| Stage [a] | I | 1/24 (4.17%) | 1 (100%) | 0 (0%) | 0.6003 |
| | II | 12/24 (54.17%) | 4 (33.33%) | 8 (66.67%) | |
| | III | 5/24 (25%) | 2 (40%) | 3 (60%) | |
| | IV | 4/24 (16.67%) | 2 (50%) | 2 (50%) | |
| Histology | Classical | 8/25 (32%) | 3 (42.86%) | 4 (57.14%) | 0.5110 |
| | Nodular lymphocyte predominant | 2/25 (8%) | 1 (100%) | 0 (0%) | |
| | Nodular sclerosing | 14/25 (56%) | 5 (33.33%) | 9 (66.67%) | |
| | Nodular sclerosing/Mixed cellularity overlap | 1/25 (4%) | 0 (0%) | 1 (100%) | |
| Bulky disease [a] | No | 21/24 (87.5%) | 7 (35%) | 13 (65%) | 0.6791 |
| | Yes | 3/24 (12.5%) | 2 (66.67%) | 1 (33.33%) | |
| History of lung disease | No | 23/25 (92%) | 7 (33.33%) | 14 (66.67%) | 0.2767 |
| | Yes | 2/25 (8%) | 2 (100%) | 0 (0%) | |
| History of lung radiation | No | 21/25 (84%) | 8 (42.11%) | 11 (57.89%) | 0.9414 |
| | Yes | 4/25 (16%) | 1 (25%) | 3 (75%) | |
| Prior G-CSF exposure [a] | No | 18/21 (85.71%) | 7 (38.89%) | 11 (61.11%) | 1 |
| | Yes | 3/21 (14.29%) | 1 (33.33%) | 2 (66.67%) | |
| Steroid use [a] | No | 1/4 (25%) | 1 (100%) | 0 (0%) | - |
| | Yes | 3/4 (75%) | 3 (100%) | 0 (0%) | |
| Age, years | mean (SD) | 25/25 (100%) | 36.11 (15.41) | 33.07 (12) | 0.6235 |
| Albumin, g/dL | mean (SD) | 25/25 (100%) | 3.92 (0.51) | 3.97 (0.6) | 0.8572 |
| Hemoglobin, g/dL | mean (SD) | 25/25 (100%) | 10.87 (2.34) | 13.29 (2.58) | 0.0370 * |
| ESR, mm/h | mean (SD) | 25/25 (100%) | 63.62 (31.03) | 30.18 (27.44) | 0.0289 * |
| ALC, /µL | mean (SD) | 25/25 (100%) | 1042 (556.39) | 1449.89 (785.02) | 0.1798 |

[a] Data missing for some patients as indicated in the total number. * denotes *p* < 0.05. ALC—Absolute lymphocyte count; BPT—Bleomycin pulmonary toxicity; ESR—Erythrocyte sedimentation rate; G-CSF—Growth factor colony stimulating factor; SD—Standard deviation.

In our patient cohort, disease characteristics, including gender, HL stage, histology, history of lung radiation, prior G-CSF, or steroid exposure, could not predict the significantly high risk of BPT (Table 4). Both patients with prior history of lung disease had clinical BPT; however, the correlation was not statistically significant. Similar association was noted with bulky disease but with $p > 0.05$.

Mean hemoglobin and erythrocyte sedimentation rate (ESR) at diagnosis were significantly associated with clinical BPT (Table 4). Patients in clinical BPT group had lower hemoglobin (mean 10.9 vs. 13.3 g/dL, $p = 0.037$) and higher ESR (mean 63.3 vs. 30.2 mm/h, $p = 0.0289$).

## 3. Discussion

In our study, 5/11 (45%) patients who had radiographic changes in PET/CT in the form of diffuse ground-glass opacities or focal areas of consolidation with increased FDG uptake developed clinical BPT. Our study is the one of only two studies that have evaluated the correlation between 18F-FDG-PET/CT findings and the onset of BPT [24]. The only other study assessing the association between 18F-FDG-PET/CT findings and BPT was reported by Falay et al. [24], who performed a retrospective review of 77 HL patients with interim and end-of-treatment PET scans; bleomycin lung toxicity was identified in 13 (17%) patients, and eight of these were identified in the interim scan, whereas six were asymptomatic.

Incidence of clinical BPT in our study was similar to ones in prior reports (10–53%) [1,2]. In cohort 2, 36% of HL patients who were treated with a bleomycin-containing regimen developed clinical BPT. This is comparable to the study by Duggan et al. who reported pulmonary toxicity in 28% of patients (224 of 814 patients) treated with a bleomycin-containing regimen [25].

The median time from start of treatment with bleomycin to the onset of BPT in our study was 3 months (range 2–11 months). This is comparable to other reports, that is, 4.2 months (range 1.2–8.2 months) reported by O'Sullivan et al. and 5 months reported by Stamatoullas et al. [7,26].

Prior studies have suggested that age > 40 years (HR 2.3, 95% CI 1.2–4.1) [1,7] and stage IV disease at presentation (HR 2.6, 95% CI 1.2–4.1) are prominent risk factors for BPT [7]. However, other studies have not been able to validate these results [26,27]. We did not observe a statistically significant correlation between BPT and risks factors, such as old age, prior exposure to radiation, history of lung disease, and HL stage at diagnosis. This could potentially be due to our small sample size.

It is known that the incidence of BPT is higher with higher cumulative dose of bleomycin, increasing to 17% for doses higher than 550 units [8]. In our cohort, the average cumulative bleomycin dose received was 154 units. This was similar to Iacovino et al., and they described two cases of fatal BPT in patients who received 100 and 165 units of bleomycin [28]. However, there have been rare reports of BPT even with lower doses (e.g., 30 units) [29].

Concomitant administration of G-CSF and bleomycin has been postulated to increase the risk of BPT [30]. In our study, only one out of three patients who received G-CSF developed BPT. This synergistic effect was significant in animal studies [31]; however, the data in humans to further understand this association are conflicting with many failing to decipher the link [1,26,27,30,32–35].

Thoracic irradiation, previously or simultaneously administered with bleomycin, has also been speculated as a risk factor for BPT [9,36]. Higher number of ABVD cycles in patients receiving chest irradiation exhibited significant correlation with BPT [36]. One study showed that the risk of BPT with consolidative RT is low if there is at least a four-week interval between chemotherapy sessions, and the RT is given after chemotherapy [37]. Concomitant use of some chemotherapeutic agent, such as gemcitabine, can also potentially increase risk of BPT [37].

Increases in inflammatory cytokines play a prominent role in the pathogenesis of BPT [12]. In our cohort, elevated ESR along with low hemoglobin were significantly related to BPT ($p$ = 0.03, 0.02 respectively). Only a few studies have reported similar correlations. One assessed ESR as a potential early marker of pulmonary toxicity from bleomycin [38]. Others have exhibited neutrophil-to-lymphocyte ratio to correlate with BPT [39]. Further studies are needed.

Our study is subject to the same limitations as all other studies that rely on retrospective review of data including limited details of treatment of BPT, unknown temporality of imaging and development of clinical bleomycin toxicity (toxicity may exist at the time of imaging). We blinded the radiologists to clinical information to offset this. Additionally, we had a small sample size, which limited some of our observations.

## 4. Materials and Methods

### 4.1. Data Collection

This retrospective cohort study was approved by institutional review board (IRB) at University of Cincinnati, IRB# UC-14-7398. Pathology records were searched to identify all non-pregnant adult HL patients (age > 18 years) diagnosed at the University of Cincinnati Medical Center (UCMC) from 2012 to 2016. Electronic medical record (EMR), Epic, was subsequently queried for these patients to identify those treated with ABVD.

The primary objective of this study was to determine whether 18F-FDG-PET/CT findings correlate with the onset of clinical BPT. 18F-FDG-PET/CT scans of HL patients treated with ABVD were evaluated for signs of pulmonary toxicity and then compared to the clinical development of BPT through EMR review (cohort 1 sample size (n) = 11). Patients with lung tumor involvement or with evidence of underlying pulmonary disease or infection on imaging were excluded. The images were evaluated by a radiology resident and a fellowship-trained nuclear radiologist who were blinded to the clinical information. For each patient, the CT and PET images were first evaluated qualitatively to assess for signs of active lung disease, such as infection or underlying chronic lung disease (both being exclusion criteria). The PET images were then qualitatively analyzed for signs of hypermetabolic processes, such as infection or tumor involvement within the lungs (also exclusion criteria). The PET images were also evaluated semi-quantitatively by the determination of the SUV$_{max}$ of the lungs. PET/CT fusion imaging was used to correlate the findings of the two concurrent imaging modalities.

A separate analysis was undertaken for all HL patients treated with ABVD from 2012 to 2016, regardless of availability of 18F-FDG-PET/CT scans, to compare those who developed clinical BPT versus those who did not (cohort 2 sample size (n) = 25). Patients with incomplete records or those who did not complete therapy at UCMC were excluded from the study. This cohort was used to evaluate secondary objective of this study—assess the correlation of clinical BPT with history of lung disease, lung radiation (before or simultaneous with bleomycin), prior G-CSF, or steroid use. We also evaluated if patient or disease characteristics, such as gender, age, ECOG PS, HL stage, histology, and bulky disease along with albumin level, hemoglobin, ESR, or absolute lymphocyte count (ALC) at the time of HL diagnosis were predictive of development of clinical BPT.

Clinical BPT was defined as acute or subacute onset of pulmonary symptoms or physical examination findings (e.g., dyspnea, cough, chest pain, and crackles) and decline in DLCO by 25% in the absence of infection, radiation-related changes, or pulmonary involvement of underlying malignancy [11,40,41].

### 4.2. Statistical Analysis

For descriptive analysis, numerical variables, such age, albumin, hemoglobin, ESR, and ALC, were summarized using median and range. Categorical variables, such as gender, stage, histology, bulky disease, and B symptoms, were reported using the frequency tables. Relationships between continuous clinical characteristics and BPT were assessed using t-tests and analysis of variance (ANOVA). Relationships between categorical variables

and BPT were assessed using Chi-Square tests. Correlation of mean $SUV_{max}$ and clinical BPT was analyzed using Mann–Whitney U test. All *p*-values were two-sided, and $p < 0.05$ was considered statistically significant. All point estimates are given at specified time points with 95% confidence intervals (CI) for the estimates. Data were analyzed using SAS 9.4 software.

## 5. Conclusions

Pulmonary findings of 18F-FDG-PET/CT imaging correlate with the development of clinical BPT in HL patients treated with ABVD therapy and can be useful for early detection and monitoring of this toxicity. Any increase in lung uptake observed in 18F-FDG-PET/CT imaging whether focal or diffuse during ABVD therapy should be considered suspicious for toxicity, which can allow early intervention to mitigate the toxicity. Additionally, low hemoglobin and high ESR can be laboratory predictors of clinical BPT; however, this needs to be validated by future studies.

**Author Contributions:** H.S., Z.O., J.S., B.M., M.P.M. and T.L. contributed substantially to the study design and the writing of the manuscript. H.S. and Z.O. drafted the manuscript, while J.S., B.M., M.P.M. and T.L. reviewed the content. R.A.J. and H.S. completed the data analysis and interpretation. All authors have read and agreed to the published version of the manuscript.

**Funding:** This research received no external funding.

**Institutional Review Board Statement:** This study was approved by institutional review board (IRB) at University of Cincinnati, IRB# UC-14-7398.

**Informed Consent Statement:** Patient consent was waived due to the retrospective nature of the study, and the study had minimal risk to the subjects.

**Data Availability Statement:** Not applicable.

**Conflicts of Interest:** The authors declare no conflict of interest.

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
