# Peer review of "FDG PET/CT as a Tool for Early Detection of Bleomycin-Induced Pulmonary Toxicity"

_2813-3307, doi:10.3390/lymphatics1010006_

Round 1

Reviewer 1 Report

Valued Colleagues,

I read your paper ”FDG PET/CT as a Tool for Early Detection of Bleomycin Induced Pulmonary Toxicity” with interest, and think it is a compelling observation about the value of commonly existing  imaging data in this disease for early detection of a dangerous treatment toxicity, that should be investigated further. However, I have several aspects that in my opinion should be improved before the article is fit for publication in Lymphatics:

1.       Abstract:

a.       Please specify that increased FDG-Uptake was measured in the lungs and the nature of the PET-scan (follow up?)

2.       Introduction:

a.       Please clarify on the combined toxicity of bleomycin and radiotherapy; “units” is not a conventional measurement, but dose information should be given in Gy per irradiated volume

b.       More detailed information about the diagnostic workup and therapy for BPT would benefit this article

3.       Results and Discussion

a.       This section should be better structured since the evaluation of two cohorts (11 patients with FDG/PET and 26 patients with ABVD-treatment) is very confusing. Did the other patients not get PET/CT scans? Please explain.

b.       Please also explain the context of the PET/CTs made (FU/re-staging). A comparison to baseline-data from initial staging would also be beneficial or should at least be discussed. As should be the decision to exclude staging and diagnostic imaging. The time of onset of changes in PET/CT during/after treatment should also be elaborated on.

c.       Since the threshold for diagnosis of BPT remains unclear, it is hard to understand why one patient had increased uptake and impaired pulmonary function, but was not classified as BPT (since it would fit the definition given under “methods”).

d.       Table 1 should give additional information, otherwise a table at this point seems unnecessary. Maybe expand the information in the likes of table 2.

e.       The pattern of FDG-uptake is not explained. An image would be helpful.

f.        What does Figure 1 show? The significance of a single mark on the graph is not given, and the x-axis needs marking.

g.       Table 2: more information about the ECOG PS of almost half the patients should be given, not only ECOG 0 selectively. Also, information about disease location is needed.

h.       Risk Factors: “history of lung radiation” needs to be more detailed – is this RT during HL-treatment? Dose(range)? Thoracal radiation is very common since HL oftentimes involves this area, so the low count of patients with history of radiation should be explained. The role of RT as a confounder should be featured more in the discussion. The extend of Bleomycin given during therapy (number of cycles, dose) should also be considered.

i.         2/2 patients with history of lung disease developed BPT, but this did not translate to being a risk factor?

j.         Steroid use is categorized into yes and no, but only shows results for 4 patients overall?

k.       Further information about the cause, treatment and severity of your BPT cases in context of your findings would be interesting.

l.         In general, the very low number of patients in this cohort limits the value of this paper immensely and should at least be compensated with detailed observations to be considered in future publications with larger numbers.

I am looking forward to your improvements and send my best regards.

Author Response

We thank the esteemed reviewer for taking the time to go through our manucript and sharing their feedback. We have tried to address their comments to our best capacity.

  1. Abstract: 
    a, Please specify that increased FDG-Uptake was measured in the lungs and the nature of the PET-scan (follow up?) We have included the suggested comments in the abstract.
  2. Introduction:
    a, Please clarify on the combined toxicity of bleomycin and radiotherapy; “units” is not a conventional measurement, but dose information should be given in Gy per irradiated volume. We agree with the reviewer. This error has been corrected.
    b, More detailed information about the diagnostic workup and therapy for BPT would benefit this article. Further description about diagnostic workup and management of BPT has been added per the suggestion.
  3. Results and Discussion
    a, This section should be better structured since the evaluation of two cohorts (11 patients with FDG/PET and 26 patients with ABVD-treatment) is very confusing. Did the other patients not get PET/CT scans? Please explain. Because we had small sample size of patients with PET scans available, we tried to add more value to this manuscript by describing a separate cohort of patients to assess the risk factors of clinical BPT. We realize how this can be confusing. Hence, we have tried to simplify this by using terms ‘Cohort 1’ (n=11) and Cohort 2’ (n=25) in methods and results.
    b, Please also explain the context of the PET/CTs made (FU/re-staging). A comparison to baseline-data from initial staging would also be beneficial or should at least be discussed. As should be the decision to exclude staging and diagnostic imaging. The time of onset of changes in PET/CT during/after treatment should also be elaborated on. The context of scans has been provided. The comparison to initial scans was made as shown in Figure 1 and 2. The decision to exclude staging and diagnostic imaging has been detailed. We do not have time of onset of changes in PET/CT during/after treatment because we focused more on PET scans and less on clinical information.
    c, Since the threshold for diagnosis of BPT remains unclear, it is hard to understand why one patient had increased uptake and impaired pulmonary function, but was not classified as BPT (since it would fit the definition given under “methods”). As detailed in the methods, we classified clinical BPT as acute or subacute onset of pulmonary symptoms or physical examination findings. This patient who had radiographic changes with FDG uptake did not have any respiratory symptoms/signs or 25% decrease in DLCO to qualify as clinical BPT. They had only minor changes in PFT findings (excluding DLCO drop of 25% or more)
    d, Table 1 should give additional information, otherwise a table at this point seems unnecessary. Maybe expand the information in the likes of table 2. We agree with this comment. However, we have limited clinical data available for cohort 1 as we focused more on PET scans and less on clinical information, nd this was collected in past. We have tried to expand Table 1 with a few more details.
    e, The pattern of FDG-uptake is not explained. An image would be helpful. We have included images with pattern of FDG uptake for 2 cases.
    f, What does Figure 1 show? The significance of a single mark on the graph is not given, and the x-axis needs marking. The description of Figure 1 (now Figure 3) is given. X axis does not have a significance here as the chart represents number of observations regardless of time correlation.
    g, Table 2: more information about the ECOG PS of almost half the patients should be given, not only ECOG 0 selectively. Also, information about disease location is needed. Additional data about ECOG PS has been added, including number of patients that had ECOG PS 0 and 1. We do not have information about disease location available. We did not collect this as we did not think this would strongly impact the objective of this study.
    h, Risk Factors: “history of lung radiation” needs to be more detailed – is this RT during HL-treatment? Dose(range)? Thoracal radiation is very common since HL oftentimes involves this area, so the low count of patients with history of radiation should be explained. The role of RT as a confounder should be featured more in the discussion. The extend of Bleomycin given during therapy (number of cycles, dose) should also be considered. We have indicated in the methods that thoracic irradiation included both prior to or simultaneous with bleomycin. Because the total number of patients in this group was low, we did not separate them for analysis. On recommendation of the reviewer, we have expanded the role of RT as a confounder in the discussion.
    i, 2/2 patients with history of lung disease developed BPT, but this did not translate to being a risk factor? That is correct. Both patients with prior history of lung disease had clinical BPT, however the correlation was not statistically significant. This is likely because 33% patients without prior history of lung disease also developed clinical BPT.
    j, Steroid use is categorized into yes and no, but only shows results for 4 patients overall? Data missing for the rest of the patients. This has been clarified in the table.
    k, Further information about the cause, treatment and severity of your BPT cases in context of your findings would be interesting. In section 2.3, details about the severity and management of BPT is indicated. We have expanded on this.
    l, In general, the very low number of patients in this cohort limits the value of this paper immensely and should at least be compensated with detailed observations to be considered in future publications with larger numbers. We agree with this. However, data on this subject is limited and any retrospective studies can add to the literature. We do feel this adds value to the literature as there is only one other manuscript published on using FDG PET to assess BPT. We have made changes as recommended by the reviewer to address reasonable concerns that were raised. We hope these changes will make the manuscript easy to understand for the reader.

Reviewer 2 Report

In this stdy, Shaikh et al describes an Integrated 18F-FDG-PET/CT scan as a tool for early detection of BPT in HL patients with very limited sample size.

Here I have some comments to this study.

1. Title: I believe you need to add ...Hodgkin Lymphoma in title at the end.

2. In line 18 you mention about 58 PET/CT scans and in the line 68 they are 54. Please clarify and if 4 PETs were excluded, then also in the line 18 change it to 54.

3. Please add to line 51 Integrated 18F-FDG-PET/CT... and also in other lines where you mention about FDG-PET/CT

4. Please provide representative images for PET/CT scans of thorax with increased SUVmax

5. Please provide a Figure with Study flow chart with all 25 Patients with inclusion, exclusion informations, and where easy to see how many of them received PET/CT scans.

6. Please provide in Materials and Methods sample size of study with exclusions.

7.  Line 148: Typo -was- written incorrectly two times.

Author Response

We thank the reviewer for taking the time to go through our study. Below are the responses to the comments of the reviewer,

  1. Title: I believe you need to add ...Hodgkin Lymphoma in title at the end. While bleomycin is mostly used for Hodgkin lymphoma, it is also used for germs cell tumors and some other diseases. Although our study population included only Hodgkin lymphoma patients, we do not have any reason to believe that our results would not apply to bleomycin used in the setting of other types of cancers. Hence, we decided to not add type of cancer in the title.
  2. In line 18 you mention about58 PET/CTscans and in the line 68 they are 54. Please clarify and if 4 PETs were excluded, then also in the line 18 change it to 54. Thank you for pointing this out. This typing error has now been corrected. Total scans were 54.
  3. Please add to line 51Integrated 18F-FDG-PET/CT... and also in other lines where you mention about FDG-PET/CT. Thank you for the recommendation. This error has now been rectified.
  4. Please provide representative images for PET/CT scans of thorax with increased SUVmax. We have provided images of PET/CT of 2 cases with increased SUVmax.
  5. Please provide a Figure with Study flow chart with all 25 Patients with inclusion, exclusion information, and where easy to see how many of them received PET/CT scans. Because we had small sample size of patients with PET scans available (patient n=11), we tried to add more value to this manuscript by describing a separate cohort of HL patients (n=25) to assess the risk factors of clinical BPT. We realize how this can be confusing. Hence, we have tried to simplify this by using terms ‘Cohort 1’ and Cohort 2’ in methods and results. If we include a flow chart, it will need to be done for both cohorts, and we worry that it might lengthen the manuscript without providing greater value.
  6. Please provide in Materials and Methods sample size of study with exclusions. Agreed. Sample size with exclusions has been detailed in materials and methods.
  7.  Line 148: Typo -was-written incorrectly two times. Thank you for pointing this out. This typing error has now been corrected.

Reviewer 3 Report

We thank the authors for their contribution "FDG PET/CT as a Tool for Early Detection of Bleomycin Induced Pulmonary Toxicity" which is interesting. Nevertheless, I do not think that data from 11 patients only is able to support the findings given. Patient numbers has to be increased considerably. Importantly, there is no information on the exact number of ABVD cycles, RT field size or RT dose which is essential for estimating the hazard ratio for pulmonary toxicity. A thorough revision of the manuscript is essential before valid conclusions can be drawn.

Author Response

We thank the authors for their contribution "FDG PET/CT as a Tool for Early Detection of Bleomycin Induced Pulmonary Toxicity" which is interesting. Nevertheless, I do not think that data from 11 patients only is able to support the findings given. Patient numbers has to be increased considerably. Importantly, there is no information on the exact number of ABVD cycles, RT field size or RT dose which is essential for estimating the hazard ratio for pulmonary toxicity. A thorough revision of the manuscript is essential before valid conclusions can be drawn. We realize that we have a small sample size of patients with PET scans available (‘Cohort 1’), however we do feel this adds value to the literature as there is only one other manuscript published on using FDG PET to assess BPT. We tried to add more value to this manuscript by describing a separate cohort of patients (n=25, ‘Cohort 2’) to assess the risk factors of clinical BPT. While this has been evaluated by many researchers, results have been inconsistent, and more data is always required to consolidate such observations as bleomycin continues to be used in many malignancies. Our study consolidated many such observations which we feel this adds to the literature. It also highlighted an observation not commonly thought of, that is, elevated ESR along with low hemoglobin were significantly related to BPT (P=0.03, 0.02 respectively). Hence, we do think our study adds value.

Reviewer 4 Report

The Authors have performed a retrospective evaluation of the role of FDG PET-CT in the early detection of bleomicine pulmonary toxicity (BPT) on patients with Hodgkin lymphoma. Particularly, they found that patients who developed clinical BPT exhibited a higher mean SUVmax in the lungs than those who did not; besides, lower hemoglobin and higher ESR, possibly suggesting the role of inflammatory cytokines in this process, were associated with a significantly higher risk of developing clinical BPT.

The manuscript is well written and results are well presented, although the small sample size, as correctly precised by the Authors. A prospective study on a wider population on the same topic would be welcome in the future. 

Best regards

Author Response

The Authors have performed a retrospective evaluation of the role of FDG PET-CT in the early detection of bleomicine pulmonary toxicity (BPT) on patients with Hodgkin lymphoma. Particularly, they found that patients who developed clinical BPT exhibited a higher mean SUVmax in the lungs than those who did not; besides, lower hemoglobin and higher ESR, possibly suggesting the role of inflammatory cytokines in this process, were associated with a significantly higher risk of developing clinical BPT.

The manuscript is well written and results are well presented, although the small sample size, as correctly precised by the Authors. A prospective study on a wider population on the same topic would be welcome in the future. 

We thank the reviewer for the comments. We do feel this adds value to the literature as there is only one other manuscript published on using FDG PET to assess BPT. We agree that prospective data with a larger sample size is required to consolidate our findings. We hope our article will shed light on such a need.

Round 2

Reviewer 1 Report

Valued colleagues,

Thank you for your improvements on the article, which were substantial in my opinion.

However, some points still need to be adressed before publication of the paper in lymphatics can be greenlighted:

- Im afraid, I must insist that the treatment regimes of the evaluated patients are explained in more detail. The role of RT in the patients of cohort 1 (dose, location, extend) has to be clarified as well as the numer of chemotherapy cycles. 

- Please check again, which unit is correct in the introduction. In general  "Gy" is the correct unit for radiation dose (hence my comment), but 300 Gy is clearly wrong, since this would be lethal. Check the references, if maybe 300cGy might be correct, BUT also include a reference to the dose (maximum dose? Dose limited to a certain volumen like V20 or V30?). 

- Please check the footnote for image 3

- I think, SUVmax values are switched in table 1. Please Check

Kind regards.

Author Response

We humbly thank the reviewer for sharing their views about the manuscript and offering recommendations. Below are our responses to their comments.

- I'm afraid, I must insist that the treatment regimes of the evaluated patients are explained in more detail. The role of RT in the patients of cohort 1 (dose, location, extend) has to be clarified as well as the number of chemotherapy cycles. We have included te details of RT and number of chemotherapy cycles.

- Please check again, which unit is correct in the introduction. In general  "Gy" is the correct unit for radiation dose (hence my comment), but 300 Gy is clearly wrong, since this would be lethal. Check the references, if maybe 300cGy might be correct, BUT also include a reference to the dose (maximum dose? Dose limited to a certain volumen like V20 or V30?). You are right about this. We have removed this detail. There is stronger data with a correlation of BPT with bleomycin dose and interval between thoracic irradiation and bleomycin administration than radiation dose.

- Please check the footnote for image 3 Thank you for highlighting this. The footnote has been revised.

- I think, SUVmax values are switched in table 1. Please Check. Thank you for pointing this out. This error has been corrected.

Reviewer 2 Report

I have no further comments to this manuscript.

Author Response

We thank the reviewer for their valuable time.

Reviewer 3 Report

I can understand that it is not possible to extend the patient collective within one week. However, i do not see that my other aspects mentioned ( exact number of ABVD cycles, RT field size or RT dose) have been adressed. I therefore can not agree upon publication.

Author Response

We thank the reviewer for taking out the time to review the manuscript and provide their valuable feedback. We have included the details of RT and the number of chemotherapy cycles.

Round 3

Reviewer 1 Report

We thank the authors for revising the manuscript. All points were addressed sufficiently. However, the overall small cohort size remains a major limitation. Other than increasing the data pool (which we understand is difficult), we do not have further recommendations. 

Reviewer 3 Report

Looking at the details of RT provided in Table 1, I have doubts on the findings, especially concerning Patient 5. 10.5 Gy is an insufficient dose for Hodgkin lymphoma. In addition, whole lung irradiation is an untypical treatment scheme for HL which is not done routinely. A direct comparison to RT to adjacent areas with only partial lung exposure is difficult. Summarizing these few patients within one gorup will not lead to solid conclusions. Overall, without data on the lung dose exposure, no estimation of the impact of RT is possible.

These biases can only avoided in a larger study with more patients.